# Analysis of Dam Inflow Variation Using the Hydrological Sensitivity Method in a Trans-Boundary River Basin: Case Study in the Korean Peninsula

**Sang Ug Kim** [1,*] and **Xiao Yu** [2]

1 Department of Civil Engineering, Kangwon National University, Chuncheon 24341, Korea
2 Engineering School of Sustainable Infrastructure and Environment, University of Florida, Gainesville, FL 32611, USA; xiao.yu@essie.ufl.edu
* Correspondence: sukim70@kangwon.ac.kr; Tel.: +82-33-250-6233

**Abstract:** Water resource planning in a trans-boundary river basin is complex because of different institutional and scientific concerns and it may become increasingly difficult as a consequence of water scarcity caused by climate change. The analysis of discharge variations in a trans-boundary river basin is very important because the results can be key to resolve complex problems including decreased hydropower generation, degraded water quality, and deficient water supplies. Despite its importance, there are relatively few studies dealing with hydrological variation in a trans-boundary river basin. Therefore, this study used the hydrological sensitivity method to identify the discharge variation in the Hwacheon dam upper basin, a representative trans-boundary river basin between South Korea and North Korea. This particular basin was selected because the inflow into the Hwacheon dam in South Korea has decreased significantly after the construction of the Imnam dam in North Korea in 2000. The hydrological sensitivity method is a simple approach to analyze variations in discharge. After collecting 51 years (1967–2017) of rainfall and inflow data, a change point that represents an abrupt change in the time series was detected by using moving average, double-mass curve analysis, Pettitt's test, and Bayesian change-point analysis. The change point detected by these methods was 1999. The hydrological sensitivity method using five Budyko-based functions was applied to a time series divided into before and after the detected change point. The average decrease after 1999 was 464.91 mm/y (or $1899 \times 10^6$ m$^3$/y). Also, the maximum and minimum decreases after 1999 were 515.24 mm/y (or $2105 \times 10^6$ m$^3$/y) and 435.32 mm/y (or $1778 \times 10^6$ m$^3$/y), respectively. Because of the increase in rainfall and the decrease in inflow since 2000, the values determined in this study are slightly larger than those from conventional studies. Finally, it is suggested that the results from this study can be used effectively to establish reasonable water resource planning in the trans-boundary river basin between South Korea and North Korea.

**Keywords:** trans-boundary river basin; hydrological sensitivity; change point; dam construction

## 1. Introduction

A trans-boundary river crosses at least one national border, either within a nation or internationally. There are about 286 trans-boundary river basins in 151 countries of more than 2.8 billion people; there are also around 22,000 km$^3$ of river discharge each year in these areas [1]. Water resources planning and management in trans-boundary river basins are generally likely to be more complex due to the different institutional, social, scientific, and political contexts involved. In addition, it is getting increasingly difficult because climate change has increased water stress, causing flood, drought, and water quality degradation. Cooley and Gleick [2] suggested that the integration of information on

future hydro-climatological conditions into complex political systems of trans-boundary river basin management can play an important role in agreements on water. Coulibaly et al. [3] evaluated the impact of climate change on water resources in the trans-boundary river basin in West Africa and showed a reduction in future annual discharge.

Therefore, there is a consensus that cooperative water resources planning and management in a trans-boundary river basin can provide opportunities to increase water benefits [4–8]. Phillips et al. [4] suggested a policy context for shared responsibility and an institutional framework for analyzing shared benefits resulting from the corporation between countries. Grey and Sadoff [5] defined the water security as the availability of water of an acceptable quantity and quality and suggested that the poorest water-insecure countries face a far greater challenge than those that have achieved water security in the last century. Leb [6] analyzed the dynamics of international cooperation and described the role of international law in achieving mutually beneficial relations. Jalilov et al. [7] and Arjoon et al. [8] used hydro-economic models to estimate the total economic benefits using various scenarios in trans-boundary river basin operations and provided an experimental case study on how the concept of benefit sharing could be used to motivate riparian countries to enhance their cooperation and develop shared water resources.

On the Korean peninsula, the trans-boundary river basin problem between South Korea and North Korea has been urgently dealt with as an important issue. The countries share two trans-boundary river basins, the Bukhangang River basin and the Imjingang River basin. The total area of the Bukhangang River is 11,343 km$^2$, 69% of which belongs to South Korea, and the rest to North Korea. The total area of the Imjingang River basin is 8897 km$^2$, 42% of which belongs to South Korea, and the rest to North Korea. In the case of the Bukhangang River basin, North Korea constructed the 121.5 m high Imnam dam in 2003, which could seriously impact drought, water quality, and hydropower generation in the downstream area in South Korea.

The construction of an upstream dam has enormous impacts on a river, especially on hydrology, sediment transport, and geomorphology. It has been reported that the variations in the discharge usually cause significant effects on low flow for irrigation, water quality, and hydropower generation [9–13]. Therefore, the construction of a dam in a trans-boundary river basin should be treated with caution because it can lead to conflicts between countries. The downstream hydrological impact of dam construction needs to be monitored and analyzed. Despite its importance, most studies [4–8] related to water problems in trans-boundary river basins have been performed mainly on the institutional or the economic aspects, and there are relatively few studies related to the hydrological changes or variations in trans-boundary river basins.

The most important procedure in analyzing the hydrological variation upstream and downstream of a basin is to quantitatively separate natural factors such as climate change and anthropogenic factors such as land-use change or dam construction. Previous studies analyzing the factors affecting hydrological variations mainly focused on the relationship between the two factors [14–16]. The studies on hydrological variation due to natural factors [17–19] dealt mainly with climate change, and similar studies on anthropogenic factors dealt mainly with gradual or abrupt change from land-use change or dam construction [20,21]. These studies generally used complex rainfall–runoff models such as VIC-3L (Variable Infiltration Capcity-3 Layer), SIMHYD (SIMHYDrology), SWAT (Soil and Water Assessment Tool) because this approach could produce reasonable results. However, Shrestha et al. [22] suggested that the ability of the rainfall–runoff models to simultaneously replicate different components of the hydrograph was not clear because of the complexity in the calibration procedure. However, recently, Pumo et al. [23] showed the interaction between climate and land use and identified the fundamental hydrological dynamics through hydrological modeling. Especially, Arnone [24] identified the effects of climate change on urban growth by the simulation of a physically based distributed hydrological model. Therefore, hydrological modeling could become an effective tool to identify natural factors and anthropogenic factors if it was well developed and calibrated.

Meanwhile, a water balance equation approach based on the hydrological sensitivity method was used to quantitatively separate the relative impact of discharge [25–29]. This approach, proposed by Budyko, analyzed the first-order effects of changes using an assumption that the change in storage can be neglected in a long period of over 10 years [30]. This approach can be used effectively to investigate long-term water balance on a larger scale without developing complex rainfall–runoff models that include calibrating and validating the parameters used.

Therefore, in this study, the discharge variation in South Korea (which is downstream of the Bukhangang River basin), before and after the construction of the Imnam dam in North Korea (which is upstream of the Bukhangang River basin), was analyzed using a water balance equation approach based on the hydrological sensitivity method. The non-parametric Mann–Kendall trend test was first used to identify the gradual trend in the collected rainfall and discharge data sets. Double-mass curve analysis, Pettitt's test, and Bayesian change point analysis were also used to detect the change point after which the change was abrupt. Finally, the results from the hydrological sensitivity method used with five different types of Budyko-based functions were compared to validate the results of this study.

## 2. Theoretical Background

### 2.1. Trend Test and Detection of Change Point

Descriptive statistics such as mean, standard deviation, and gradual and abrupt changes in trends in acquired hydro-meteorological data provide important clues to determine how appropriate the final result is. The Mann–Kendall non-parametric trend test [31,32] is the most widely used method to capture the gradual trend in a time series [33,34]. When $x_i$ and $x_k$ in a time series $X = [x_1, x_2, \ldots, x_n]$ are independent, the statistic $S$ is first calculated using Equation (1). Then, the statistical hypothesis test is performed to find the $p$-value using $Z_{MK}$ in Equation (2) under the selected significance level $\alpha$:

$$S = \sum_{k=1}^{n-1} \sum_{i=k+1}^{n} sgn(x_i - x_k), \ \text{sgn}(x_i - x_k) = \begin{cases} 1 & if \ (x_i - x_k) > 0 \\ 0 & if \ (x_i - x_k) = 0 \\ -1 & if \ (x_i - x_k) < 0 \end{cases} \tag{1}$$

$$Z_{MK} = \begin{cases} (S-1)/\sqrt{Var[S]} & if \ S > 0 \\ 0 & if \ S = 0 \\ (S+1)/\sqrt{Var[S]} & if \ S < 0 \end{cases} \tag{2}$$

The results of the abrupt trend test are more important than those of the gradual trend test because they provide the information needed to determine the change point that represents the abruptly changed location on the time series. At first, the moving average method can be used to capture the abrupt change. This method analyzes data points by creating series of averages of different subsets of the whole data series and is commonly used to smooth out short-term fluctuations and highlight long-term trends. However, this method can produce different results depending on the time lag selected for the moving window. Therefore, this subjective determination can impact the final result. Second, the double-mass curve can be used to roughly detect the change point [29]. It is a graph that draws the accumulated values of one variable against those of other. A change in the gradient of this curve indicates that the statistical relationship between two variables changed abruptly.

A few statistical tests, such as Pettitt's test and Bayesian change point analysis, can be used to detect the change point more objectively. Pettitt's test [35] is a type of non-parametric trend test that can detect a change point. In this test, the null hypothesis is $H_0$: some variables $T$ follow one or more distributions that have the same location parameter (indicating no change). The alternative hypothesis is $H_A$: a change point exists in the time series. The non-parametric statistics used by

Pettitt are defined in Equation (3). The change point is detected at $K_T$, provided that the statistics are significant. The probable significance of $K_T$ is approximated in Equation (4):

$$K_T = \max|U_{t,T}|, \ U_{t,T} = \sum_{i=1}^{t} \sum_{j=t+1}^{T} sgn(x_i - x_j) \tag{3}$$

$$p \approx 2 \exp\left(\frac{-6K_T^2}{T^3 + T^2}\right) \tag{4}$$

Another objective change point test is the Bayesian change point (BCP) analysis, and its advantage is the detection of many change points by simultaneous analysis of multiple change points. Let us assume that $\{x_i\}_{i=1}^{i=n}$ is a sequence of an observed time series with the probability density functions $p_1(x), p_2(x), \ldots, p_n(x)$. For a sequence of $n$ independent random variables $\mathbf{X} = (X_1, X_2, \ldots, X_n)$, the change point model can be described by Equation (5), in which the parameters $\theta_1 \neq \theta_2 \neq, \ldots, \neq \theta_n$ in the probability density functions and the change points $\tau_1, \tau_2, \ldots, \tau_{n-1}$ are unknown. Therefore, this time series can be divided into $n$ homogenous groups if the locations of change points are determined.

$$X_i \sim \begin{cases} p_1(x) = p(x_i|\theta_1), & 1 \leq i \leq \tau_1 \\ p_2(x) = p(x_i|\theta_2), & \tau_1 < i \leq \tau_2 \\ \quad \vdots & \quad \vdots \\ p_n(x) = p(x_i|\theta_n), & \tau_{n-1} < i \leq \tau_n \end{cases} \tag{5}$$

Barry and Hartigan [36] used the product partition model, which assumes that the probability of any partition is proportional to a product of prior cohesions. Recently, the BCP analysis was applied to detect the abrupt change in hydrological or meteorological time series [37–39]. The BCP package developed by Erdman and Emerson [40] for the software R was used to perform BCP analysis in this study. In this algorithm, the posterior mean and the change of probability are calculated from the MCMC (Markov Chain Monte Carlo) technique. In the MCMC algorithm of the product partition model, the implementation is performed by Equation (6):

$$\frac{p_i}{p_i - 1} = \frac{P(U_i = 1 \,|\mathbf{X}, U_j, j \neq i)}{P(U_i = 0 \,|\mathbf{X}, U_j, j \neq i)} = \frac{\left[\int_0^{p_0} p^b (1-p)^{n-b-1} dp\right] \left[\int_0^{\omega_0} \frac{\omega^{b/2}}{(W_1 + B_1 \omega)^{(n-1)/2}} d\omega\right]}{\left[\int_0^{p_0} p^{b-1} (1-p)^{n-b} dp\right] \left[\int_0^{\omega_0} \frac{\omega^{(b-1)/2}}{(W_0 + B_0 \omega)^{(n-1)/2}} d\omega\right]} \tag{6}$$

where $p$ and $\omega$ are random variables in the product partition model. Also, $b$ is the number of blocks, $W_0$ is the within-block sum of squares, and $B_0$ is the between-block sum of squares in the case of $U_i = 0$. Also, $W_1$ and $B_1$ are similarly defined in the case of $U_i = 1$.

In this study, the moving average method and the double-mass curve analysis were first applied to roughly detect the range of change points roughly, and then Pettitt's test and BCP analysis were applied to determine an accurate change point.

## 2.2. Framework for Separating Natural and Anthropogenic Factors

The runoff processes in a specific basin can be changed by various factors in the hydrologic cycle, such as rainfall, temperature, infiltration, and evapotranspiration, in addition to human activities such as land-use change, installation of intake facilities, afforestation or deforestation, and dam construction. However, these factors can be simply regrouped into natural and anthropogenic factors. Therefore, the runoff discharge can be modeled as in Equation (7):

$$Q = f(N, A) \tag{7}$$

where $Q$ is the discharge in a specific location, $N$ and $A$ are variables representing natural factors and anthropogenic factors, respectively, and $f$ is a function representing the relationship between two factors. After a simple application of a first-order approximation to Equation (7), Equation (8), which represents the variation of discharge, can be derived. In a real case, the natural factors and the anthropogenic factors may be correlated. However, it is assumed that the two variables are independent of each other in this study.

$$\Delta Q = \frac{\partial Q}{\partial N} \Delta N + \frac{\partial Q}{\partial A} \Delta A = \Delta Q_N + \Delta Q_A \tag{8}$$

where $\Delta Q$ is the total discharge variation, and $\Delta Q_N$ and $\Delta Q_A$ represent discharge variations due to natural factors and anthropogenic factors, respectively.

Discharge variations due to natural factors are likely to proceed gradually, but those due to anthropogenic factors are likely to proceed abruptly after a change point. Figure 1 represents the total discharge variation before and after a change point (decreasing case). The natural period is defined as that impacted by only the natural factor, and the impacted period is defined as that impacted by the natural and the anthropogenic factors. In Figure 1, $\overline{Q}_{obs}^1$ and $\overline{Q}_{obs}^2$ are the averages of observed discharge in two periods, and $\overline{Q}_{sim}^2$ is the average of simulated discharge without the anthropogenic factor in the impacted period. A hydrological sensitivity method or a hydrological model should be used to simulate $\overline{Q}_{sim}^2$. In Equation (8), when $\Delta Q_A < 0$, it can be calculated by Equation (9):

$$\Delta Q_A = \overline{Q}_{obs}^2 - \overline{Q}_{sim}^2 \tag{9}$$

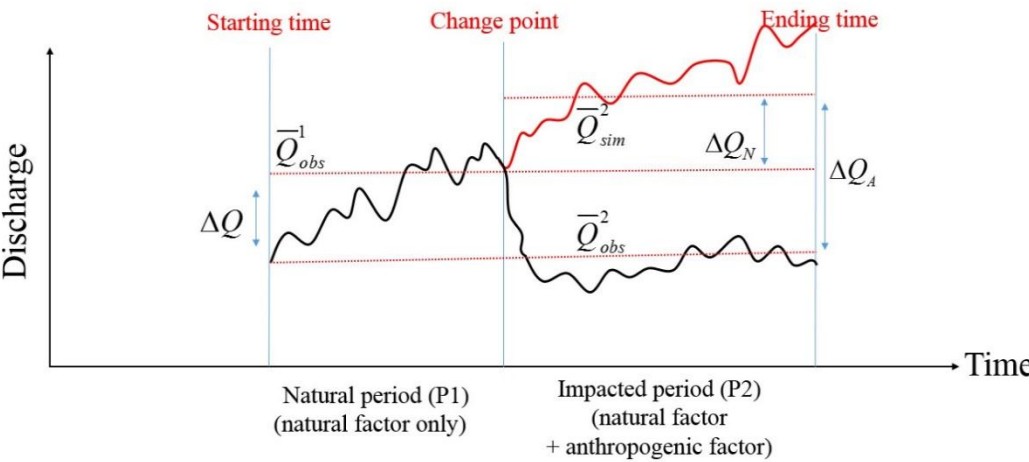

**Figure 1.** Schematic diagram to separate natural and anthropogenic factors.

The procedure used in this study to separate the variations due to natural and anthropogenic factors from the total variations in the discharge can be performed through the following steps:

(Step 1) Detect the change point and separate periods P1, P2

(Step 2) Calculate the observed discharge $\overline{Q}_{obs}^1$ and $\overline{Q}_{obs}^2$ in periods P1, P2;

(Step 3) Calculate the total variation of discharge: $\Delta Q = \overline{Q}_{obs}^1 - \overline{Q}_{obs}^2$

(Step 4) Estimate $\Delta Q_N$ using the hydrological sensitivity method (or hydrological model)

(Step 5) Calculate $\Delta Q_A$ in Equations (8) and (9)

### 2.3. Hydrological Sensitivity Method

Hydrological sensitivity is the change in the mean annual discharge that occurs in response to the change in the mean annual precipitation and potential evapotranspiration (*PET*). Water balance is described using discharge, precipitation, and actual evapotranspiration (*AET*) by Equation (10):

$$P = Q + AET + \Delta S \tag{10}$$

where *P*, *Q*, and *AET* are precipitation, discharge, and actual evapotranspiration during a period, respectively. $\Delta S$ is the change in storage, but it can be neglected since its sum can be assumed to be zero in a long period of over 10 years [41].

Furthermore, the energy balance for a basin described in Equation (11) should be considered for use in hydrological sensitivity methods:

$$N = L \times AET + H + \Delta G \tag{11}$$

where, *N*, *L*, *H*, and $\Delta G$ are net radiation, latent heat for evaporation, sensible heat flux during a period, and the change of net ground heat flux, respectively. Also, $\Delta G$ can be assumed as zero for the same reason as $\Delta S$ [41]. After eliminating $\Delta S$ and $\Delta G$ in Equations (10) and (11), it is assumed that *PET* = *N/L* and $\gamma = H/(L \times AET)$. Equation (12) can be derived simply by dividing Equation (11) by Equation (10):

$$\frac{PET}{P} = \frac{AET}{P} + \frac{AET \times \gamma}{P} = \frac{AET}{P}(1 + \gamma) = \phi \tag{12}$$

where $\gamma$ and $\phi$ are called Bowen ratio and aridity index, respectively. Because the Bowen ratio can be assumed as a function of the aridity index [41], $\gamma = f(\phi)$, Equation (12) can be finally rearranged into Equation (13):

$$\frac{AET}{P} = \frac{\phi}{1 + f(\phi)} = F(\phi) \tag{13}$$

where $F(\phi)$ is the Budyko curve function.

Many studies have been performed to find $F(\phi)$ using actual evapotranspiration and precipitation time series. The five commonly used functions are reported in studies by Schreiber [42], Ol'dekop [43], Pike [14], Budyko [30], and Fu [44]. The estimated functions are described in Table 1.

**Table 1.** Commonly used Budyko functions.

| Name of Function | $F(\phi)$ |
|---|---|
| Schreiber (1904), [42] | $1 - e^{-\phi}$ |
| Ol'dekop (1911), [43] | $\phi \tanh(1/\phi)$ |
| Budyko (1948), [30] | $[\phi \tanh(1/\phi)(1 - e^{-\phi})]^{0.5}$ |
| Pike (1964), [14] | $(1 + \phi^{-2})^{-0.5}$ |
| Fu (1981), [44] | $1 + \phi - (1 + \phi^{2.5})^{1/2.5}$ |

In this study, the aridity index $\phi$ was calculated by Equation (12), where *PET* was estimated by the Penman–Monteith method. After calculating $\phi$ in the basin of interest, *AET* in Equation (13) can be calculated using the five Budyko functions given in Table 1. Meanwhile, considering Equation (10), the variation in precipitation can be described as $\Delta P = \Delta Q + \Delta AET$. $\Delta Q$ can be assumed to represent $\Delta Q_N$ in the hydrological sensitivity approach because in the long term, $\Delta Q$ is impacted only by natural factors. Therefore, the variations in the discharge due to natural factors can be written as in Equation (14):

$$\Delta Q_N = \Delta P - \Delta AET \tag{14}$$

Koster and Suarez [45] developed Equation (14), representing the relationship between *AET* and the Budyko functions using Equations (12)–(14). The details of how Equation (15) was derived can be found in a previous study [29].

$$\Delta Q_N = \frac{\Delta P}{P} Q \left(1 + \frac{\phi F'(\phi)}{1 - F(\phi)}\right) + \frac{\Delta PET}{PET} Q \left(-\frac{\phi F'(\phi)}{1 - F(\phi)}\right) = \varepsilon_P \frac{\Delta P}{P} Q + \varepsilon_{PET} \frac{\Delta PET}{PET} Q \qquad (15)$$

where $\varepsilon_P$ and $\varepsilon_{PET}$ are elasticity coefficients of precipitation and potential evapotranspiration, and $\varepsilon_P + \varepsilon_{PET} = 1$. $F'(\phi)$ is the derivative of the Budyko curve function, $F(\phi)$. Equation (15) can only be used to estimate the variations in discharge due to natural factors, $\Delta Q_N$; the discharge variation due only to anthropogenic factors, $\Delta Q_A$, can be estimated using Equation (9).

## 3. Study Area and Data Characteristics

### 3.1. Selection of the Study Area

The study area for this research was the Hwacheon dam upper basin, a part of the Bukhangang River basin (Figure 2). This basin was selected because there have been continuous and accurate observations of how the construction of the Imanm dam in North Korea impacts the inflow into the Hwacheon dam. The Hwacheon dam upper basin (4084.92 km$^2$) consist of three sub-basins, the Imnam dam sub-basin (1008, 2384.68 km$^2$), the Pyeonghwaui dam sub-basin (1009, 940.45 km$^2$), and a part of the Chuncheon dam sub-basin (a part of 1010, 759.79 km$^2$).

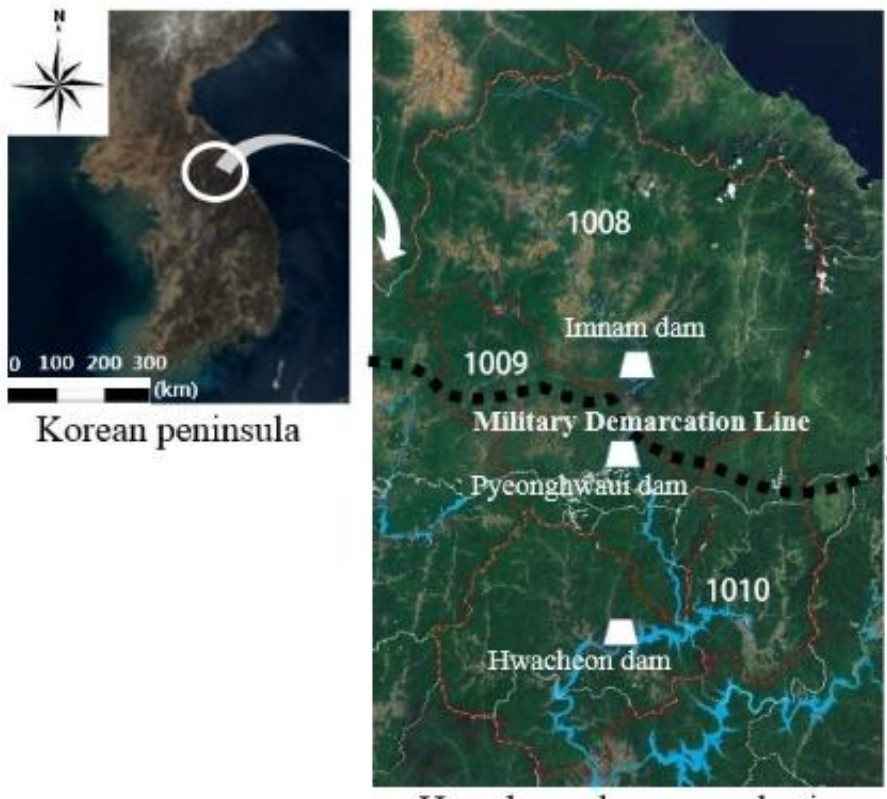

**Figure 2.** Selected basin and three sub-basins. (Dotted black line: Military demarcation line, Dotted red line: Border line of sub-basins).

The Imnam dam (height of 121.5 m, width of 710 m, and total storage of 2600 × 10$^6$ m$^3$) in sub-basin 1008 was completed in 2000. The dam generates hydropower by conveying water from sub-basin 1008 to the East Sea. However, the conveying discharge is not accurately known.

The Pyeonghwaui dam (height of 125.0 m, width of 601 m, and total storage of $2630 \times 10^6$ m$^3$) in sub-basin 1009 was constructed in 2005.

However, the effect of this dam on the regulation of the Hwacheon dam cab be negligible under in the long term because this dam outflows the entire inflow from the Imnam dam. The Hwacheon dam (height of 81.5 m, width of 435 m, and total storage of $1018 \times 10^6$ m$^3$) in sub-basin 1010 was constructed in 1944 and has been operating mainly to generate hydropower.

### 3.2. Data Collection and Characteristics

The annual average rainfall and annual average inflow data were collected from 1967 to 2017 (51 years) at the Hwacheon rainfall and dam stage gauges. Furthermore, the annual average inflow data (m$^3$) for the 51 years were converted to annual average inflow depth data (mm) by dividing them by the area of the Hwacheon dam upper basin (4084.92 km$^2$). Figure 3 shows the collected annual average rainfall and inflow depth data. The red dotted line and the black dotted line are the results of the moving average with a 10-year moving window. The annual average rainfall series show a gradual decrease until around 1997 and an increase since 1998, except for the period from 2014 to 2017 when there was a severe meteorological drought in South Korea. However, it shows that the annual average inflow depth has been decreasing steadily despite the increase of rainfall after 1998. This suggests, intuitively, that the hydrological response in terms of discharge changed around 1997. It is also reasonable to suggest that this change might have occurred as a result of the regulation of the Imnam dam, which was constructed in 2000. It also shows that the annual average rainfall series increased abruptly around 1998, but it is difficult to decide whether the result of the 10-year moving average method on the annual average inflow depth series showed an abrupt change or not. Table 2 shows the descriptive statistics, including mean and standard deviation, of the annual average rainfall and inflow depth series. The values of skewness and kurtosis were almost the same. It can be suggested that the annual average inflow should be about 760.1 mm because the approximate runoff coefficient in South Korea is known to be about 0.6. However, in Table 2, the difference of the mean between the two data series was high because the annual average inflow (592.7 mm) was smaller than the estimated value (760.1 mm).

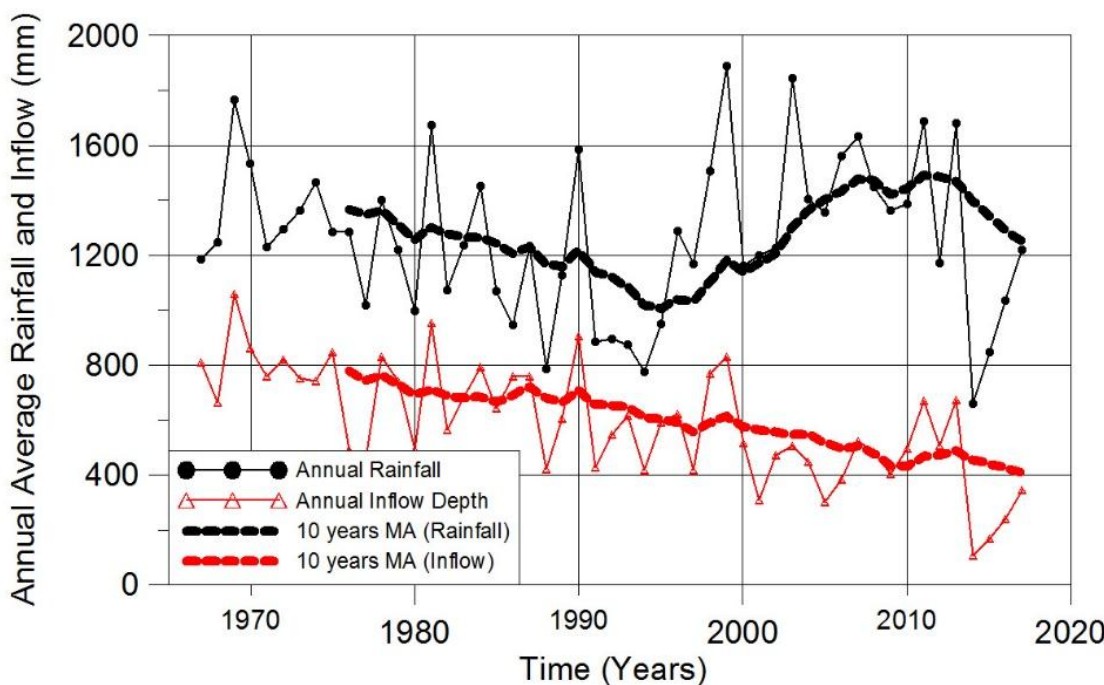

**Figure 3.** Annual average rainfall and inflow and moving average (MA) lines.

**Table 2.** Statistical characteristics of the two collected data series.

| Data | Mean (mm) | Standard Deviation (mm) | Skewness | Kurtosis |
| --- | --- | --- | --- | --- |
| Annual average rainfall | 1266.8 | 286.6 | 0.099 | −0.391 |
| Annual average inflow | 592.7 | 206.8 | −0.092 | −0.335 |

## 4. Application and Results

### 4.1. Results of Change-Point Detection

Although the change point in this study can be determined directly using prior information from the data, a few statistical tests were performed to detect it accurately. The gradual change was first analyzed using the Mann–Kendall non-parametric trend test before the change point was detected by the abrupt difference between the two collected data sets. Table 3 shows the results of the Mann–Kendall test at the 5% significance level. The result for the annual average rainfall series shows that there was no gradual change in the trend, but the result for the annual average inflow depth data show a statistically significant trend.

**Table 3.** Results of Mann–Kendall non-parametric trend test (significance level: 5%).

| Data | *p*-Value | Decision |
| --- | --- | --- |
| Annual average rainfall | 0.8718 | No gradual trend |
| Annual average inflow | $1.6525 \times 10^{-6}$ | Gradual trend |

The analyses were performed using the double-mass curve (with cumulative rainfall and inflow depth data), Pettitt's test, and BCP analysis to detect a reasonable change point in these data series. Figure 4 shows the double-mass curve using the cumulative annual average rainfall and the cumulative annual average inflow depth from 1967 to 2017. The slopes of the curve from 1967 to 1997 and from 1998 to 2017 are different. Therefore, it can be suggested that the change point occurred between 1997 and 2000.

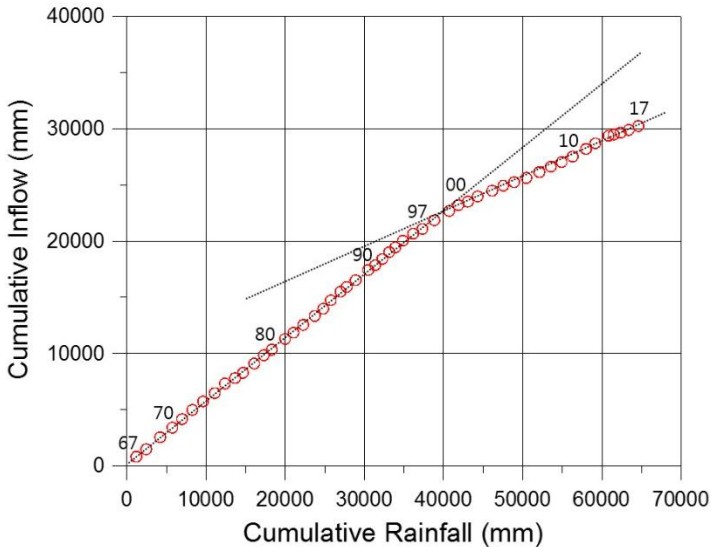

**Figure 4.** Double-mass curve to detect the change point.

After determining the range of the change point, Pettitt's test and BCP analysis using a package in R determined the temporal location of the change point more accurately. Table 4 and Figure 5 show the result of Pettitt's test and the BCP analysis at the 5% significance level. In Table 4, the abrupt change

point was not detected in the annual average rainfall series; however, the change point was detected in 1999 in the annual average inflow depth series by Pettitt's test. This, therefore, suggests that the statistical characteristics of these series are different before and after 1999.

**Table 4.** Results of Pettitt's test (significance level: 5%).

| Data | *p*-Value | Decision |
| --- | --- | --- |
| Annual average rainfall | 0.4032 | No change point |
| Annual average inflow | $1.1284 \times 10^{-8}$ | Change point: 1999 |

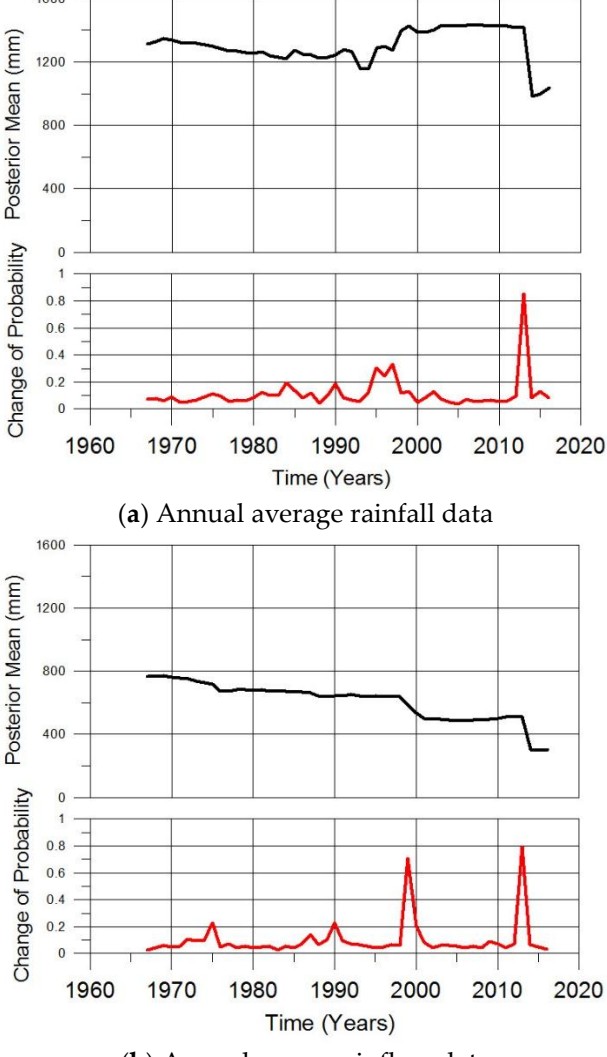

(**a**) Annual average rainfall data

(**b**) Annual average inflow data

**Figure 5.** Results of Bayesian change-point analysis.

Figure 5a,b represent the posterior mean (upper panel) and change in probability (lower panel) of the annual average rainfall and inflow depth data. The posterior mean is the value of the posterior probability of the product partition model by Barry and Hartigan [36]. The change in probability represents the probability that an abrupt change can occur at some point. Generally, when the change in probability is higher than 0.6–0.7, the temporal location of this value can be determined as a change point. The maximum value of the change in probability in the annual average rainfall data, in the lower panel of Figure 5a, was 0.337, except for 2014. A severe drought occurred in 2014, and this year was excluded in this study because this point was not a change point due to the Imnam dam construction. Therefore, it is clear that there was no change point in the annual average rainfall data.

However, the maximum value of the former in the annual average inflow depth data, in the lower panel of Figure 5b, was 0.705 in 1999. This suggests that the statistical characteristics of the annual average inflow depth data changed after 1999.

Finally, the change point in the annual average inflow depth data was determined to be 1999 because Pettitt's test and the BCP analysis showed the same results. This suggests that the change point in the annual average inflow depth data occurred only because of the construction of the Imnam dam, since the anthropogenic factors such as land use did not change significantly for a decade in the Hwacheon dam upper basin. At sub-basin 1010, the urban area was 10.89 km$^2$ in 1985, and 11.12 km$^2$ in 2016. Although this study cannot investigate the land-use change at sub-basin 1008 in North Korea, it can be suggested that the effect of land-use change in North Korea was too weak to change the results.

### 4.2. Results of the Hydrological Sensitivity Method

The change point in the annual average inflow time series was determined as 1999 using the double-mass curve, Pettitt's test, and BCP analysis. Therefore, the annual average rainfall and inflow depth data for the 51 years are separated into two series, 1967–1998 and 1999–2017. After the separation at the change point (1999), the hydrological sensitivity method was applied to calculate variations in inflow depth due to natural factors ($\Delta Q_N$) and anthropogenic factors ($\Delta Q_A$).

Figure 6 shows the detailed procedure used to calculate $\Delta Q_N$, $\Delta Q_A$, and $\Delta Q$ as a flow chart. The relative humidity, wind speed, temperatures (min and max), and solar radiation data were collected at Chuncheon meteorological station, and the Penman–Monteith method was applied. The annual average rainfall for the 51 years was 1357 mm, and the annual average potential evapotranspiration (*PET*) for the same period was 802.5 mm. The value of $\phi$ was 0.591342. After calculating *PET* in Step 1, Step 2 to Step 6 were performed to calculate the variations in inflow depth, $\Delta Q_N$, $\Delta Q_A$, and $\Delta Q$. Table 5 shows the final results.

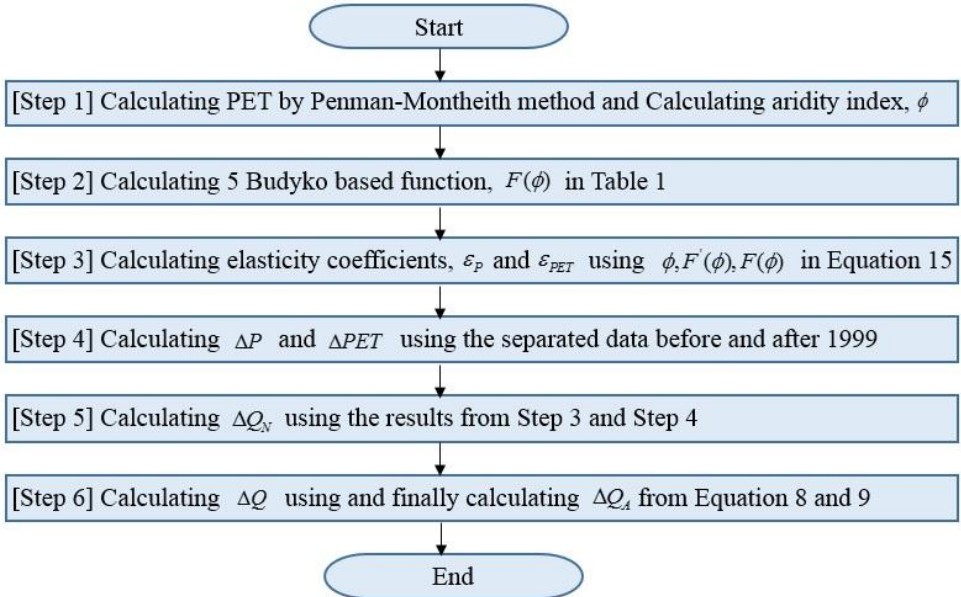

**Figure 6.** Flow chart for calculating the inflow variation by two factors. *PET*: potential evapotranspiration.

In Table 5, the inflow depth from natural factors increased the least when the Schreiber function was used and the most when the Budyko function was used. This suggests that the range of the increased inflow depth due to natural factors in the Hwacheon dam upper basin was from 194.22 mm/y to 274.15 mm/y before and after the change point. Furthermore, the inflow depth from anthropogenic factors decreased the least when the Schreiber function was used and the most when the Budyko function was used. This suggests that the decrease of the inflow depth due to anthropogenic factors, such as the construction of the Imnam dam, was in the range between 435.31 mm/y and 515.24 mm/y.

Finally, this study suggests that the final decreased inflow depth due to the construction of the Imnam dam was 464.91 mm/y, which is the average of five Budyko-based functions. This value was converted to $1899 \times 10^6$ m$^3$/y by multiplying it by the area of the Hwacheon dam upper basin. Furthermore, this value was compared to the results of conventional studies; the five Budyko-based functions ranged from $1778 \times 10^6$ m$^3$/y (using the Schreiber function) to $2105 \times 10^6$ m$^3$/y (using the Budyko function).

**Table 5.** Results of the hydrological sensitivity method.

| Functions | $F(\phi)$ | $\varepsilon_P$ | $\varepsilon_{PET}$ | $\Delta P$ (mm/y) | $\Delta PET$ (mm/y) | $\Delta Q_N$ (mm/y) | $\Delta Q$ (mm/y) | $\Delta Q_A$ (mm/y) |
|---|---|---|---|---|---|---|---|---|
| Schreiber (1904), [42] | 0.4464 | 1.5913 | 0.5913 | 406.30 | 49.55 | 194.22 | 241.09 | 435.31 |
| Ol'dekop (1911), [43] | 0.5525 | 1.9505 | 0.9505 | 406.30 | 49.55 | 231.90 | 241.09 | 472.99 |
| Budyko (1948), [30] | 0.4966 | 2.3532 | 1.3532 | 406.30 | 49.55 | 274.15 | 241.09 | 515.24 |
| Pike (1964), [14] | 0.5090 | 1.7681 | 0.7681 | 406.30 | 49.55 | 212.76 | 241.09 | 453.85 |
| Fu (1981), [44] | 0.4914 | 1.7044 | 0.7044 | 406.30 | 49.55 | 206.08 | 241.09 | 447.17 |

Lim [46] developed the SSARR (Streamflow Synthesis And Reservoir Regulation [47]) rainfall–runoff model to calculate the decrease in inflow due to the construction of Imnam dam. Lim suggested that the inflow decreased to $1600 \times 10^6$ m$^3$/y. Ahn et al. [48] developed the rainfall–runoff model SWAT-K (Soil and Water Assessment Tool-Korea, [49]) and the water balance model K-MODSIM (K-MODelling and SIMulation [50]). They calculated the decreased inflow as $1220–1620 \times 10^6$ m$^3$/y. Therefore, the values obtained in this study are higher than those from conventional studies. It can be estimated that these results are due to the increased rainfall and decreased inflow since 2000.

## 5. Conclusions

The accurate and continuous monitoring and analysis of discharge variations is fundamental to establish a reasonable water resources plan in a trans-boundary river basin. This is because the characteristics analyzed can help to resolve complex institutional, political, and scientific issues between stakeholders. The results of the analysis of discharge variations can provide important data to establish plans for hydropower generation, water quality control, and irrigation, because discharge variations have a great effect on all of these. Therefore, this study focused on identifying discharge variations instead of economic and social problems in a trans-boundary river basin.

Conventional studies usually use complex rainfall–runoff models to simulate discharge variations. However, this study used a hydrological sensitivity method, a simpler approach than the rainfall–runoff modelling, to estimate dam inflow variations in the Hwacheon dam upper basin. This basin is a representative trans-boundary river basin between South Korea and North Korea and is especially important because the inflow has been decreasing downstream, in South Korea, after the construction of Imnam dam upstream, in North Korea in 2000.

After collecting data such as rainfall and inflow depth from a 51-year period (1967–2017), the change point representing an abrupt change in the data was detected in 1999 using the moving average, double-mass curve, Pettitt's test, and BCP analysis. The hydrological sensitivity method was performed on two time series, separated into before and after the detected change point, using five different Budyko-based functions. Finally, it was suggested that the range of the decreased inflow due to anthropogenic factors, specifically, the construction of the Imnam dam, was from 435.32 mm/y (or $1778 \times 10^6$ m$^3$/y) to 515.24 mm/y (or $2105 \times 10^6$ m$^3$/y). The average of the five Budyko functions was 464.91 mm/y (or $1899 \times 10^6$ m$^3$/y). When this value was compared with the results from the

conventional studies, the decreased variation in inflow from this study was larger than those from the conventional studies because of the increased rainfall and decreased inflow discharge after 2000.

　　The results of this study can be used to analyze hydropower generation, water quality degradation, and management of the irrigation water supply in South Korea. The results of this study can play an important role in establishing a sustainable water resources management in a trans-boundary river basin between South Korea and North Korea.

　　However, the limitations of this study should be addressed in future work. Firstly, it is necessary to develop a hydrological model including North Korean's anthropogenic factors such as dam regulation, land use, and river construction. Secondly, the uncertainties in the five Budyko-based functions used should be assessed further with a Budyko-based function that suits the Korean hydrological conditions. Finally, the yearly time scale should be changed to a seasonal or monthly time scale to provide more detailed temporal variations in inflow; also, the evaporation on the reservoir surface needs to be evaluated.

**Author Contributions:** S.U.K. established the research direction and performed the analysis; X.Y. provided constructive suggestions and suggested the limitations of this study.

**Funding:** This study was performed with a grant from the Basic Science Research Program through the National Research Foundation of Korea funded by the Ministry of Education (2018R1D1A1B07040409) and with a research grant of Kangwon National University in 2018 (visiting scholar program).

**Acknowledgments:** We thanks for three anonymous reviewers who gave us constructive comments to improve this manuscript.

**Conflicts of Interest:** The authors declare no conflict of interest.

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
