# Peer review of "Analysis of Dam Inflow Variation Using the Hydrological Sensitivity Method in a Trans-Boundary River Basin: Case Study in the Korean Peninsula"

_water, doi:10.3390/w11020395_

Round 1
Reviewer 1 Report
This study proposes to evaluate the hydrological sensitivity method to identify the discharge variation in the Hwacheon dam upper basin, a representative trans‐boundary river basin between South Korea and North Korea. This basin was selected because the hydrology response into the Hwacheon basin in South Korea has decreased significantly after the construction of the Imnam dam in North Korea. The authors used the hydrological sensitivity method on a period of 51 years (1967~2017) of rainfall and inflow data.
That said, I think there are few issues to address. I recommend publication contingent on revisions as described in the attached pdf file in the form of tracks of the manuscript.

Author Response
Dear Reviewer 1
Thanks for your comments to improve this manuscript. We revised the draft version of the manuscript as the comments from three reviewers. The attached pdf file is reply to Reviewer 1. You can find replies to each comments. See the memos in pdf file.
Best.
Authors.

Reviewer 2 Report
Dear authors,
I made some comments regarding your paper. Please, read the attached file.
Best regards,
Your reviewer

Author Response
Dear Reviewer 2
Thanks for your comments to improve this manuscript. We revised the draft version of the manuscript as the comments from three reviewers. The attached file is reply to Reviewer 2. You can find replies to each comments.
Best.
Authors.

Reviewer 3 Report
The manuscript deals with the detection of changes in hydrological time-series due to the Imnam dam construction (North Korea) in the case of the transboundary Bukhangang River basin at the Hwacheon dam (South Korea). The topic is of interest for the scientific community; the paper is well written and organized. I have only some concerns, discussed below, mainly related with the presentation of the work.
The Authors perform first different statistical tests for trend or abrupt change detection (Mann-Kendall, Pettitt, double mass, etc.) based on a series of annual mean discharge of 51 years. Results suggest that an abrupt change occurred around 1999, when the dam was finished to built in 2000; hence, prior knowledge confirm statistical test results. Indeed, the separation of the two periods could have been done solely based on prior knowledge and the visual inspection of the data.
Second, the Authors apply the Budyko equation to compute the “natural” and anthropogenic components of the overall variation of the inflow discharge at the Hwacheon dam. The proposed approach allows reconstruct the effect of the upstream reservoir on the downstream one, without knowing the regulation rules of the upstream dam; indeed, it is only known that the Imnam dam generates hydropower by conveying water to the East Sea. The proposed approach relies on the assumption of separating precipitation, temperature evapotranspiration and infiltration (which are changing due to natural factors accoding to Authors framework), from those factor that are of anthropogenic nature and act at the “local” scale, such as riverworks construction. Human interventions such as deforestation or, in general, land use changes (urbanization), cannot be detected by using this kind of approach, simply because they affect infiltration, evapotranspiration and locally also temperature. Further, the method applies only in the specific case dam regulation implies to convey water discharge out of the catchment. I would like to see this issue discussed in the manuscript.
Further, I would like to have more details on alternative but less straightforward methods for dam regulation detection and on their limitations.
Finally, I personally believe that the series of data used in the analysis is very short to clearly detect the statistical behavior of the observed discharge, yet this is strictly not necessary in this specific case. In figure 3, I can see that both rainfall and discharge are decreasing during the period about 1970 to 1995; perhaps if we look back to previous periods this behavior is not observed to the same extent. In such a case, how does the method is expected to perform?
Figure 5 should be better described in the text or in its caption. In general, the Bayesian approach for multiple change point detection is poorly described in the text. Furthermore, what does the high probability value observed in both rainfall and discharge series in 2014 mean?
Author Response
Dear Reviewer 3
Thanks for your comments to improve this manuscript. We revised the draft version of the manuscript as the comments from three reviewers. The attached file is reply to Reviewer 3. You can find replies to each comments.
Best.
Authors.

Round 2
Reviewer 3 Report
The Authors have addressed all the Reviewer comments. Hence my opnion is that the manuscript can be published as it stands, apart from editorial changes.